# Validating quality standards in Palestinian emergency departments: An e-Delphi survey approach

**Abed Alr'oof Bani Odeh**[1]*, **Lee Wallis**[1], **Motasem Hamdan**[2], **Willem Stassen**[1]

**1** Division of Emergency Medicine, University of Cape Town, Cape Town, South Africa, **2** Faculty of Public Health, Al-Quds University, East Jerusalem, Occupied Palestinian Territory

* abedsaleem@yahoo.com

**Data Availability Statement:** All relevant data are within the manuscript and its Supporting information files.

**Funding:** The author(s) received no specific funding for this work.

## Abstract

To validate Palestine's previously derived emergency department quality standards (EDQS) using an e-Delphi survey. A two-round e-Delphi survey validated the EDQS, developed in an earlier study through a literature review and consensus-building among Palestinian emergency medicine and healthcare quality experts. The study purposively sampled 53 emergency department and healthcare quality experts with over 5 years of experience. A Likert scale was used to rate the standards on readability, clarity, and comprehensiveness in the initial round to reach consensus on the EDQS, with detailed feedback. An expanded expert group refined the shortlisted standards in the next phase. Lime Survey collected data anonymously. A set of 100 EDQS was validated through a two-round e-Delphi survey. In the initial round, 103 standards were presented, and consensus was achieved, resulting in a refined list of 100 standards. Among these, 39 standards fell under the clinical pathway domain, and 61 under the administrative pathway domain. In the second round, the validity of these standards was affirmed, with 96.4% consensus for clinical standards and 97.3% for administrative standards. Additionally, seven subdomains of EDQS were associated with the clinical pathway domain: triage, treatment, transportation, medication safety, patient flow, and medical diagnostic services, and nine subdomains were linked to the administration pathway domain: documentation, information management systems, access-location, design, leadership, management, workforce staffing, training, equipment, supplies, capacity-resuscitation rooms, resources for a safe working environment, performance indicators, and patient safety-infection prevention and control programs. The study validated context emergency department quality standards in Palestine, with over 97% consensus indicating a commitment to quality care. Experts suggest further research on implementation feasibility. Validated standards can aid healthcare leaders in resource allocation, staff training, and enhancing patient care, potentially leading to significant improvements in emergency healthcare in Palestine.

**Competing interests:** The authors have declared that no competing interests exist.

## Introduction

The quality of healthcare provided in emergency departments (EDs) is extremely important, as it directly impacts patient outcomes, safety, and satisfaction [1]. In the unique context of Palestinian EDs, establishing and validating quality standards is crucial to improving healthcare delivery and patient safety.

The quality of care for individual patients is determined by their ability to access effective care to maximize health benefits while addressing their needs [2]. Among numerous definitions, the quality of healthcare for individuals is defined as "*whether individuals can access the health structures and processes of care which they need and whether the care received is effective*" [2].

In the last ten years, numerous governmental and professional bodies have developed quality measures, like indicators or standards, to enhance healthcare quality (HCQ) and pinpoint poor-quality care across structural, procedural, and outcome aspects [3]. Furthermore, six dimensions of quality care have been identified, encompassing safety, effectiveness, patient-centeredness, timeliness, efficiency, and equity [4].

Quality indicators are "*quantitative measures that provide information about the effectiveness, safety, and/or people-centeredness of care*" [5]. Quality standards, defined as "*optimum levels of performance*" [6], play a crucial role in assessing HCQ and its dimensions, including ED services [4].

An emergency healthcare system is crucial for saving lives and protecting patient well-being, making it a cornerstone of any healthcare system [7, 8]. Moreover, it is essential for universal health coverage, dealing with acute conditions in children and adults, such as injuries, infections, exacerbations of diseases, and obstetric emergencies. It's the first link to the health system for many, identifying urgent conditions, resuscitating, and referring severely ill patients while providing definitive care for others [7].

The increase in trauma and non-communicable diseases in low- and middle-income countries (LMICs) underscores the critical importance of effective emergency care. Global entities such as the World Bank, and the World Health Organization (WHO) are focusing on enhancing emergency care systems in areas with limited resources [9]. However, the progress of its development is still in its nascent phase in numerous areas [8]. Therefore, assessing the quality of emergency care delivery is essential to improving the overall quality of healthcare in this context [8].

The Palestinian healthcare setting is characteristic in its requirements and challenges [10]. It must address not only the traditional medical and clinical demands but also those imposed by the socio-political and economic conditions unique to the region. The frequent disruptions and resource limitations further complicate providing reliable, efficient, and equitable emergency healthcare in this context [11]. To improve care delivery and patient outcomes in Palestinian EDs, it's important to establish quality standards designed for this setting. Validating these standards is crucial to ensure they accurately measure what they are intended to measure [12].

The EDQS subjected to validation, were previously identified through a literature review, yielding 115 quality standards. Subsequently, a panel of local experts in Palestinian EDs and HCQ assessed these standards, resulting in a refined list of 103. These standards have been organized into two main domains and 16 sub-domains [13]. Through an e-Delphi survey, this study seeks to validate the quality standards of Palestinian EDs, aiming to enhance emergency care and patient safety in the country. The research aligns with ongoing healthcare system enhancements, highlighting the importance of standardized quality measures in EDs.

The EDQS were established through a thorough literature review to ensure they are based on solid evidence and best practices. For example, previous research identified key dimensions of emergency care, highlighting the importance of organized triage systems to improve patient flow and reduce waiting times through systematic approaches and standardized criteria that prioritize health issues. Accurate triage aims to deliver the right treatment at the right time to the right patient, making it a critical standard [14, 15]. Other studies have emphasized that effective leadership and management are crucial for coordinating and sustaining emergency services, which directly influences the quality standards of emergency care leadership [16]. Similarly, The patient safety standards, based on the WHO Patient Safety Framework, prioritize error prevention and risk management [17]. By incorporating these principles, the standards were designed to reflect the core dimensions of good care identified across diverse healthcare systems, ensuring their relevance and adaptability to the Palestinian context.

Research on quality improvement and patient safety in Eastern Mediterranean EDs, particularly in Palestine, is limited [18]. While some studies address ED worker safety, burnout, violence, and medication [18–20], the Ministry of Health (MoH) assesses infrastructure, equipment, and staff based on WHO standards [21]. However, a notable gap in validated quality standards, accreditation, or proactive use in EDs indicates a lack of validated measurement systems and standards for emergency healthcare in Palestine. The study aims to fill this gap by validating Palestinian contextual EDQS.

Conventional methods for HCQ standards sometimes depend on generic frameworks developed with limited stakeholder involvement, rendering them inappropriate for low-resource environments. Efforts to implement international emergency care standards in Palestinian healthcare have faced challenges due to poor contextual alignment. This study adopts an innovative methodology by creating EDQS customized to local requirements and thoroughly validating them using the Delphi technique and stakeholder feedback. This guarantees that the standards are both relevant and feasible for execution in resource-limited settings [13, 22].

This study aimed to validate the Palestinian standards for the quality of ED services. This supports the validity of assessment processes based on these standards, and the results can be relied upon "*True measure*" [12], and used for improvement and comparison. The involvement of experts from the public, private, and academic sectors in validating Palestinian ED standards is crucial for their successful application and long-term sustainability. This includes the support of the MoH leadership.

## Methods

### Study design

This study employed a two-round e-Delphi survey methodology to validate quality standards for EDs in Palestine. Utilizing an iterative approach, this technique aimed to foster consensus by collecting data from a carefully selected panel of experts [3, 23]. Experts with at least 5 years of experience in emergency medicine (EM), health care quality, or both, were invited to validate contextual EDQS for Palestine.

To ensure respondent anonymity, a quasi-anonymous method was implemented, allowing survey responses to remain confidential while being known to the researcher [24]. A similar approach was employed to establish quality standards [24]. The chosen methodology was considered highly suitable for achieving the research objectives. This manuscript is presented in accordance with the Conducting and Reporting Delphi Studies (CREDES) reporting standards [25], (Fig 1). Furthermore, this research was carried out with ethical approval from the Human Research Ethics Committee (HREC) at the University of Cape Town, reference number (015/2022), and approval from the Palestinian MoH (DHM220367).

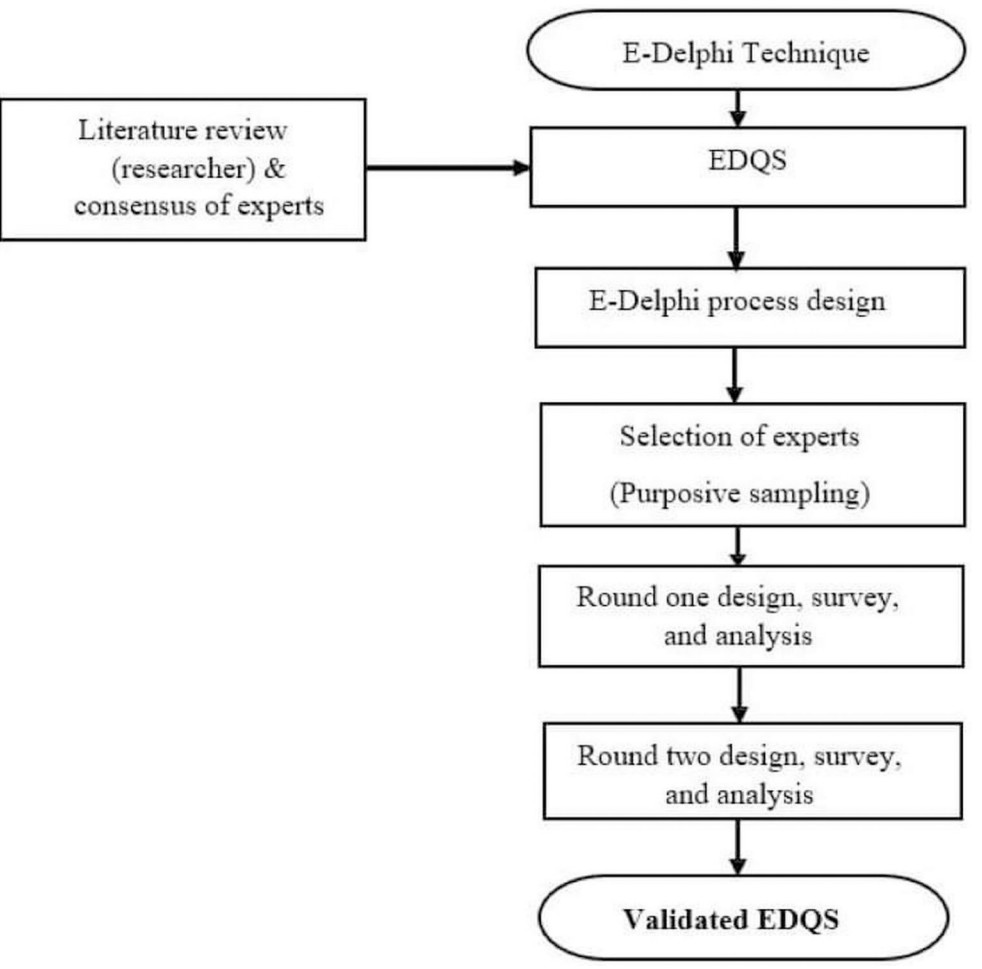

**Fig 1. Two rounds of E-Delphi process to validate EDQS.**

## Quality standards for validation

A total of 103 quality standards specific to Palestinian EDs were presented to the panel of experts for validation [26]. These standards were compiled through a combination of literature review and consensus among Palestinian experts in the field of EM and HCQ experts with more than 5 years' experience in each discipline. (Bani Odeh et al., 2024). Among these standards, 39 related to the clinical domain, while the remaining 64 fell under the administrative domain, (S1 Appendix).

## Delphi panel recruitment and sample

A total of 61 experts received invitations to participate in the e-Delphi survey. Participants received information about the study's objectives and data handling through a LimeSurvey link sent via email. They provided informed consent by completing an online Delphi Study Consent Form, which detailed the study's purpose, procedures, and potential risks, and was documented electronically on the LimeSurvey. By filling out the survey, participants approved their involvement, and all data were analyzed anonymously to ensure confidentiality. They were informed that their responses would remain confidential and used only for research purposes.

A purposive sampling method was utilized to identify experts who had actively contributed to the validation process [23, 27]. Two expert groups were selected based on their experience in hospital emergency care and HCQ at local, regional, and international levels. In the two rounds, only 53 (87%) experts completed the survey. Most of the experts who did not complete the survey apologized and provided various reasons such as busy schedules and too long a survey to complete. The experts who participated in the survey were from various institutions and organizations. These included: MoH hospital EDs, private hospital EDs, doctor syndicates, nursing syndicates, and experts from various universities [23, 28].

## Study instrument

The instrument used for the validation of EDQS in this research is a two-round e-Delphi survey, designed and administered through the LimeSurvey (LimeSurvey GmbH, Hamburg, Germany) platform hosted by Cape Town University. In the first round, experts in the field of emergency care were presented with a comprehensive list of 103 candidate quality standards. These experts were asked to rate each standard's readability, clarity, and comprehensiveness in general using a Likert scale, allowing them to express their level of agreement or disagreement. The experts were also encouraged to provide detailed feedback, comments, and suggestions for further refinement, this round was conducted from 15 to 30 May 2023. In the second round, the shortlisted quality standards, which had passed the consensus threshold in the first round, were presented to the expanded expert panel including those who participated in the first round. This round aimed at further refinement, specifically focusing on three key validation criteria: readability, clarity, and comprehensiveness, this round was conducted from 15–30 October 2023. Table 1 provides clear definitions for these criteria, aiming to establish a common understanding and highlight enhancements made during the refinement process.

These three criteria have proven effective in validation studies across healthcare and align with ISQua principles for developing health standards as well. The consensus on clarity, readability, and comprehensiveness ensures that the standards are relevant, understandable, measurable, beneficial, and achievable (RUMBA) [12, 29].

The use of LimeSurvey as the data collection tool facilitated the systematic and anonymous gathering of expert opinions, ensuring that the resulting quality standards would be grounded in both consensus and expert feedback. This method was pivotal in tailoring the standards to the specific context of emergency care in Palestine [28, 30].

## Delphi rounds

**Round 1: Evaluation and shortlisting.** In the first round of the e-Delphi survey, the panel of experts received a list of 103 candidate quality standards [26]. Each expert was asked to read and assess each standard, aiming to identify the most pertinent and applicable ones for inclusion in the ultimate set of quality standards for EDs. The experts utilized a 5-point Likert scale to rate each standard, indicating their level of agreement, which ranged from "strongly agree"

**Table 1. Validation criteria definitions.**

| Criteria | Definition / Meaning |
|---|---|
| **Readability** | • **To what extent the statement of the quality standard can be read and understood?** |
| **Clarity** | • **To what extent is the statement of quality standard worded carefully to be understandable and have maximum obviousness?** |
| **Comprehensiveness** | • **To what extent does the quality standard statement provide a complete detailed and description of requirements?** |

to "strongly disagree". Additionally, experts were invited to give feedback and suggestions for improving each standard without introducing new ones [31].

Responses from the first round were reviewed and categorized, then transferred from the Lime Survey to a Statistical Package for Social Science version 26 (SPSS) for descriptive analysis (frequencies and percentages) [32]. This analysis aimed to determine consensus for each domain, subdomain, and associated standards, defined as $\geq$ 80% agreement and an interquartile range (IQR) of $\leq$ 1, where IQR measures dispersion around the median. This consensus threshold was established following Delphi study practices to ensure high reliability and minimal variability in expert evaluations [30].

Based on the feedback and comments received in the first round, specific standards underwent modifications and merging 2 standards, resulting in a renumbering of the list to 100 standards. A summary table was prepared to provide a clear overview of the expert's ratings and comments., [30] (S2 Appendix).

**Round 2: Refinement and validation.**  In the second round, experts were presented with a shortlist of 100 standards that had passed the consensus threshold in the first round [28]. Experts were asked to re-evaluate these standards, considering their readability, clarity, and comprehensiveness. They were also encouraged to provide additional feedback and suggestions for refinement [28, 30].

## Final analysis and consensus

The responses from the second round were analyzed to assess the standards' validity and to determine if there was consensus among the experts regarding their suitability for the Palestinian emergency care context. The final list of validated domains, subdomains, and associated quality standards was derived from the standards that met the consensus threshold defined at $\geq$ 80% agreement (strongly agree or agree) [3, 28, 30].

## Ethical consideration

The research was carried out with ethical approval from the Human Research Ethics Committee (HREC) at the University of Cape Town, reference number (015/2022), and approval from the Palestinian MoH (DHM220367). Participants were provided with detailed information about the study's objectives and data handling through a LimeSurvey link. The online Delphi Study Informed Consent Form enabled them to review the study details and consent to participate. Consent was indicated by reading the provided information and completing the survey, which was considered formal approval for their participation. And the data were analysed anonymously.

## Inclusivity in global research

Additional information regarding the ethical, cultural, and scientific considerations specific to inclusivity in global research is included in the Supporting Information checklist (S3 Appendix).

## Results

### Characteristics of participants

In the first round, 22 out of 26 invited experts fully participated, yielding an 85% response rate. These respondents comprised 86% local, 9% regional, and 5% international experts with backgrounds in EM, HCQ, or both, ensuring a diverse and comprehensive approach to standard validation.

**Table 2. Characteristics of the respondents in the two e-Delphi survey rounds.**

| The first round of e-Delphi survey | | | | | | | |
|---|---|---|---|---|---|---|---|
| Invited Experts number | Respondents number and % | Respondent background | | | Respondents by location | | |
| | | EM | HCQ | Both field | Local | Regional | International |
| n = 26 | n = 22, 85% | n = 8 | n = 8 | n = 6 | (n = 27), 87% | (n = 2), 6.5% | (n = 2), 6.5% |
| The second round of e-Delphi survey | | | | | | | |
| Invited Experts number | Respondents number and % | Respondent background | | | Respondents by location | | |
| | | EM | HCQ | Both field | Local | Regional | International |
| n = 35 | n = 31, 89% | n = 21 | n = 20 | n = 17 | (n = 27), 87% | (n = 2), 6.5% | (n = 2), 6.5% |

In the second round, 35 experts were invited, including 22 from the initial round and the rest were new invitees. 31 experts participated fully, resulting in an 89% response rate. Among these participants, 87% were local Palestinian healthcare experts, 6.5% were from the eastern Mediterranean region, and another 6.5% were international experts. The overall number of respondents in both rounds was 53. Table 2 presents participant characteristics, response rate, geographical distribution, and professional backgrounds in both rounds.

## Delphi results

**Results of the first round.** In the first round of the e-Delphi survey, 2 domains, 16 subdomains, and 103 associated EDQS were presented to the expert's panel for their opinions and validation of the standards (S1 Appendix).

**Consensus achievement.** In the first round of the e-Delphi survey, 103 standards were evaluated by the experts. Among these, 22 standards received a consensus rate of over 80%. However, they were slightly modified and reworded based on the recommendations of the experts. In addition, 4 standards related to workforce staffing and training subdomain were combined into one standard, with B4003 and B4006 being merged with B4001, and B7003 being merged with B7001. Moreover, one of the standards was split into two. Finally, 77 standards were agreed upon by the experts without any modifications, as listed in (S2 Appendix).

According to the results, experts have reached a consensus on the two main domains of EDQS. It was found that the clinical domain (A) standards achieved a level of consensus of 94.2%, whereas the administration domain (B) standards achieved 94.3% consensus. Furthermore, all sub-domains reached a consensus among the experts, including seven sub-domains of clinical standards (A1-A7) and nine sub-domains of administrative standards (B1-B9). Overall, 100 (94.2%) of EDQS met the cut-off point of $\geq$ 80%.

During the validation process, standards that had an average rating equal to or above the 80% consensus threshold proceeded to the next round. 100 out of the 100 EDQS met the level of agreement, along with two relevant domains and 16 subdomains (S4 Appendix).

## Results of the second round

**Shortlisted standards.** In the second round of the e-Delphi survey, 2 domains, 16 subdomains, and 100 shortlisted EDQS were moved from e-Delphi one and presented to the expert panel for validation.

**Final consensus.** In the second round of the survey, all 100 EDQS were included and achieved a consensus level of 80% or more among experts without any comments.

All 39 clinical domain standards (A) achieved a consensus level of 96.7% for readability, 96.3% for clarity, and 96.3% for comprehensiveness. There were no comments or recommendations for changes, and the overall level of consensus for this domain was 96.4%.

**Table 3. Summary of consensus results for two rounds of e-Delphi for EDQS.**

| Clinical Domain | Delphi 1 | | | Delphi 2 | | | |
|---|---|---|---|---|---|---|---|
| Subdomains | [a] % | Disagreement | Mean | [a] % | Disagreement | Mean | [a] IQR |
| A1 | 93.5 | 6.5 | 4.7 | 96.4 | 3.6 | 4.8 | ≤ 1 |
| A2 | 94.2 | 5.8 | 4.7 | 96.4 | 3.6 | 4.8 | ≤ 1 |
| A3 | 92.5 | 7.5 | 4.6 | 94.2 | 5.8 | 4.7 | ≤ 1 |
| A4 | 95.3 | 4.7 | 4.8 | 96.8 | 3.2 | 4.8 | ≤ 1 |
| A5 | 93.2 | 6.8 | 4.7 | 96.8 | 3.2 | 4.8 | ≤ 1 |
| A6 | 94.8 | 5.2 | 4.7 | 96.9 | 3.1 | 4.8 | ≤ 1 |
| A7 | 95.5 | 4.5 | 4.8 | 97.3 | 2.7 | 4.9 | ≤ 1 |
| **A** | **94.1** | **5.9** | **4.7** | **96.4** | **3.6** | **4.8** | **≤ 1** |
| Admin. Domain | Delphi 1 | | | Delphi 2 | | | |
| Subdomains | [a] % | Disagreement | Mean | [a] % | Disagreement | Mean | [a] IQR |
| B1 | 93.5 | 6.5 | 4.7 | 97.3 | 2.7 | 4.9 | ≤ 1 |
| B2 | 94.6 | 5.4 | 4.7 | 97.5 | 2.5 | 4.9 | ≤ 1 |
| B3 | 94.7 | 5.3 | 4.7 | 97.2 | 2.8 | 4.9 | ≤ 1 |
| B4 | 93.6 | 6.4 | 4.7 | 97.3 | 2.7 | 4.9 | ≤ 1 |
| B5 | 95 | 5.0 | 4.8 | 97.2 | 2.8 | 4.9 | ≤ 1 |
| B6 | 95.1 | 4.9 | 4.8 | 97.7 | 2.3 | 4.9 | ≤ 1 |
| B7 | 95.1 | 4.9 | 4.8 | 96.8 | 3.2 | 4.8 | ≤ 1 |
| B8 | 93 | 7.0 | 4.7 | 96.5 | 3.5 | 4.8 | ≤ 1 |
| B9 | 94 | 6.0 | 4.7 | 98.1 | 1.9 | 4.9 | ≤ 1 |
| **B** | **94.3** | **5.7** | **4.7** | **97.3** | **2.7** | **4.9** | **≤ 1** |

[a] Threshold of consensus ≥ 80% and IQR ≤ 1.

Furthermore, all 61 standards related to administration (B) achieved a high level of agreement, with 97.3% consensus for readability, 97.2% for clarity, and 97.3% for comprehensiveness. No comments or recommendations for modifications were received, and the overall level of consensus for this domain remained unchanged at 97.3% (S5 Appendix).

Consensus is achieved when 100 EDQS reach an agreement of 80% or more, and IQR is ≤ 1. In such cases, no additional rounds are required, and the standards are deemed valid [30], Table 3 summarizes the e-Delphi survey results, showing expert consensus for clinical and administrative domains and each sub-domain. It also displays the percentage of disagreement with the standards within the subdomains.

(S6 Appendix) presents the final list of 100 validated EDQS across two main domains—clinical and administrative—and 16 subdomains.

## Discussion

This study aimed to validate EDQS in Palestine to improve emergency services and patient safety. The final 100 standards were validated through consensus-building and expert input [28, 30]. These validated standards are essential for Palestinian EDs to monitor performance, identify areas for improvement, and enhance overall care quality [24]. Consensus through two iterative rounds of the e-Delphi survey is crucial for validating standards, as it helps minimize bias and noise, offers controlled feedback for modifying judgments, and ensures stability in standards [30, 33].

Our study revealed that most EDQS received strong support from experts, indicating their validity and robustness. Despite the high level of agreement, our panel consisted of diverse

experts with different backgrounds, expertise, and geographic locations, ensuring a comprehensive assessment. This diversity enhanced the credibility and applicability of our findings, affirming that the EDQS were thoroughly and inclusively evaluated [25].

The validated EDQS in Palestine, identified through the e-Delphi survey, aligns with existing literature on crucial aspects such as triage, patient assessment, treatment protocols, referral processes, leadership, management, and patient safety. These standards reflect the dimensions of quality care emphasized in previous studies [17, 34], which highlight the importance of comprehensive clinical, organizational, and safety considerations in emergency services. This alignment confirms the adherence of Palestinian EDQS to established quality care principles in emergency medicine, affirming their relevance in both local and global healthcare contexts [29]. Furthermore, this research supports ongoing efforts to improve emergency care in Palestine by providing context-specific standards that guide healthcare providers in enhancing service quality, addressing care gaps, and facilitating broader national healthcare reform initiatives.

However, there are also differences and innovations introduced through the study. The standards were validated with input from national, regional, and international experts and are context-specific, meaning that they are specifically designed to address the particular difficulties that the Palestinian healthcare system faces [10, 23, 31]. Unlike traditional methods of setting quality standards, this approach considers unique aspects specific to Palestine, including cultural differences and security concerns, which results in a more effective and sustainable solution for improving emergency care in the region [10, 28, 30].

Additionally, the study's focus on achieving consensus through a rigorous e-Delphi survey approach, involving multiple rounds of expert input and feedback, demonstrates a commitment to developing standards that are widely accepted and validated by a diverse group of stakeholders [23, 27, 28].

The validated EDQS in Palestine has significant implications for improving the quality of emergency care services in Palestinian EDs. By providing a benchmark for evaluating and improving ED services, these standards might help to reduce patient morbidity and mortality, improve patient satisfaction, and enhance the overall quality of healthcare services in the country. Firstly, they guide administrators, clinicians, and policymakers in Palestine to improve ED services and provide high-quality patient care. By offering clear directives, they streamline clinical processes, promoting efficiency and effectiveness [35, 36].

Secondly, adherence to these quality standards can improve patient outcomes by ensuring the use of evidence-based clinical protocols, appropriate medications, diagnostic tests, and timely patient assessment and treatment. This adherence promotes better clinical outcomes, reduces complications, enhances overall patient safety in EDs, and reduces unnecessary costs [36, 37].

Moreover, the standards contribute to enhancing overall efficiency by promoting effective leadership and management, ensuring adequate staffing, and training, and optimizing resource utilization. Addressing administrative processes, such as documentation and performance indicators, streamlines operations, improving overall emergency care efficiency. These implications underscore the potential of these standards to positively transform emergency care delivery in Palestine [16, 36].

The validated EDQS in Palestine serves practical purposes, including guiding the implementation of clinical pathways, staff training, and education. As a result, they provide a foundation for quality improvement efforts, aiding in resource allocation and facilitating performance measurement and monitoring. Additionally, integrating the standards enhances patient safety practices and initiatives aimed at improving satisfaction in emergency care [36].

The validated EDQS may face difficulties during implementation, including conflict, limited resources, and workplace violence. Violence against workers in Palestinian EDs is widespread, with younger and less experienced health workers who have direct patient contact being the most vulnerable. Additionally, frequent episodes of violent conflict with Israel hinder any improvements [19]. Strategies include cooperating with local and international partners, advocating for funding, and providing staff training. Prioritizing staff well-being, mental health support, and work-life balance are crucial. Developing user-friendly online training modules and engaging local communities in standard development are necessary. Advocating for increased healthcare funding, partnerships with relevant sectors, and impactful initiatives are essential for addressing economic challenges. Continuous senior management commitment to establishing an organizational culture dedicated to improvement, along with continuous monitoring and evaluation, is vital to overcoming these challenges [13, 36].

This study aimed to validate EDQS but did not assess their implementation or impact on the quality of care provided. Therefore, future research should explore the feasibility and acceptability of adapting the validated standards to different Palestinian EDs, as well as conducting pilot testing in Palestinian EDs. This should consider factors such as cultural differences, patient demographics, and organizational characteristics.

## Limitations

While the EDQS were previously developed using available literature, their validation was based on subjective expert opinions via the e-Delphi method. This method, although beneficial, can be influenced by individual biases and limited expertise among the panel. Although attempts were made to have a diverse expert group, differences in knowledge and viewpoints could have affected the consensus. Moreover, not including stakeholders like patients and frontline ED staff in the Delphi process may have limited the range of perspectives, potentially impacting the thoroughness of the validated EDQS. Technological issues and internet access for some participants were also challenges.

## Conclusion

This study successfully validated 100 EDQS specific to the Palestinian context. The validated standards in this study are a crucial step toward enhancing quality and patient safety practice in Palestinian EDs. Despite challenges in implementing these standards, such as insufficient resources, political instability, and frequent attacks on hospitals by the Israeli occupation, the potential benefits are significant and deserve further attention and investment.

## Supporting information

**S1 Appendix. Consensus-based quality standards for emergency departments in Palestine (CBQSEDP) presented for validation and outcome of using two rounds of e- Delphi technique.**
(DOCX)

**S2 Appendix. Consensus level and modified EDQS after e-Delphi round 1.**
(DOCX)

**S3 Appendix. Inclusivity in global research.**
(DOCX)

**S4 Appendix. EDQS validation results / e-Delphi 1.**
(DOCX)

**S5 Appendix. EDQS validation results / e-Delphi 2.**
(DOCX)

**S6 Appendix. List of validated emergency departments quality standards (EDQS)–e-Delphi.**
(DOCX)

## Author Contributions

**Conceptualization:** Abed Alr'oof Bani Odeh.

**Supervision:** Lee Wallis, Motasem Hamdan, Willem Stassen.

**Writing – review & editing:** Abed Alr'oof Bani Odeh.

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
