## [Decision Letter · Decision Letter 0]

21 Nov 2024

PONE-D-24-25666Validating Quality Standards in Palestinian Emergency Departments: An e-Delphi Survey ApproachPLOS ONE

Dear Dr. Bani Odeh,

Thank you for submitting your manuscript to PLOS ONE. After careful consideration, we feel that it has merit but does not fully meet PLOS ONE’s publication criteria as it currently stands. Therefore, we invite you to submit a revised version of the manuscript that addresses the points raised during the review process.

Kind regards,

Naeem Mubarak, PhD

Academic Editor

PLOS ONE

Journal Requirements:

Additional Editor Comments:

The manuscript has a good deal of merit for publication but needs minor revisions to further improve its quality and impact.

Reviewers' comments:

Reviewer's Responses to Questions

**Comments to the Author**

1. Is the manuscript technically sound, and do the data support the conclusions?

Reviewer #1: Yes

Reviewer #2: Yes

2. Has the statistical analysis been performed appropriately and rigorously? 

Reviewer #1: Yes

Reviewer #2: Yes

3. Have the authors made all data underlying the findings in their manuscript fully available?

Reviewer #1: Yes

Reviewer #2: Yes

4. Is the manuscript presented in an intelligible fashion and written in standard English?

Reviewer #1: Yes

Reviewer #2: Yes

5. Review Comments to the Author

Reviewer #1: Thank you for providing me the opportunity to review the manuscript entitled “Validating Quality Standards in Palestinian Emergency Departments: An e-Delphi Survey Approach” by Alr’oof Bani Odeh et al. The approach taken to validate quality standards within Palestinian emergency healthcare settings was crucial. The authors laid a strong foundation for future assessments and the implementation of improved service standards. With the involvement of a diverse panel of experts, the study effectively aimed at achieving consensus through a thorough examination of clinical and administrative domains. This comprehensive evaluation affirmed the robustness of the Emergency Department Quality Standards (EDQS).

While the manuscript fulfils its objective of laying the groundwork for assessing and validating quality standards in Palestinian Emergency Departments, I would like to offer a few minor recommendations that may further enhance the clarity and flow of your work:

1. Introduction (Elaboration on Previous Shortcomings in Literature): It would be beneficial to elaborate on the shortcomings of past researches in their attempts to better elucidate on the gap bridged by the study. ‘Unlike traditional methods of setting quality standards ..’ This statement may be supported by highlighting such previous orthodox methodologies adopted, allowing the study’s innovations to stand out more clearly. Kindly consider this addition under the introduction.

2. Methods:

a. Delphi panel recruitment and sample: In the sentence “In the two rounds only 53... and too long a survey to complete,” it is implied that the length of the survey may have introduced bias. Could you kindly elaborate on the measures taken to ensure that participants who completed the survey were not influenced by the survey's length, and how potential bias was mitigated?

b. Readability, clarity and comprehensiveness: It is recommended to provide further details on the sources utilized for determining these as the three key validation criteria. This will enhance understanding of the criteria selection and strengthen the study's methodological transparency.

c. Delphi rounds (Subheading – Round 1 Evaluation and Shortlisting) : In the line “…with consensus threshold defined at ≥ 80% agreement and IQR ≤ 1,” I suggest elaborating on the tool, instrument, or method used to determine these thresholds. Providing this additional information would aid in further validating the standards used to analyze the findings.

3. Discussion (Literature comparison): While the approach is novel it is advisable to further support your findings by comparing them to existing or past research under discussion. The authors note that "the progress of its development is still in its nascent phase" and that the research "aligns with existing literature on EDQS in several ways." By citing relevant previous studies, the authors can strengthen their arguments, provide clearer context, and reinforce the credibility of their findings. Additionally, the statement "The research aligns with ongoing healthcare system enhancements" would benefit from further elaboration, highlighting how the study contributes to current advancements in the field.

4. Literature Review sources as the Foundation of the Validation Process: The literature review serves as a critical foundation for the development of the quality standards being validated. It has been noted that "these standards were compiled through a combination of literature review" and that "the EDQS subjected to validation were previously identified through a literature review." To strengthen this validation process, it is recommended to provide specific examples of previous research that informed the standards. This would not only support the process but also clarify the criteria for selection.

5. Formatting Adjustments: To improve the professional presentation of the manuscript, I suggest the following formatting adjustments:

a. Including line numbers would facilitate easier referencing for both the author and reviewer, and using "Page X of Y" for page numbering would enhance organization and navigation.

b. Additionally, please maintain font size consistency, as the font size under the subheading “Consensus Achievement” appears to decrease from 11 to 10 in the line referencing “workforce staffing and training subdomain.” Consistent font size will aid in readability.

c. It is recommended to rename the "Background" section to "Introduction" to more accurately reflect its content. The title "Introduction" better captures the broader elements discussed, such as research gaps, objectives, and global impact, as opposed to the historical context implied by "Background."

Reviewer #2: Abstract Is too long especially methodology and conclusion

Give more idea about previously Palestine's derived emergency department quality standards in introduction

Real Trauma quality standards have no space in this eDelphi survey which should have been priority

Better is results in place of findings.

103 quality standards specific to Palestinian EDs were too excess, should have evalusated less nimber , yield would have been more prosperous and cutting edge research.

You have done eDelphi survey, need to know what new learning from your survey is for the Palestine.

6. PLOS authors have the option to publish the peer review history of their article (what does this mean?). If published, this will include your full peer review and any attached files.

Reviewer #1: No

Reviewer #2: No

---

## [Author Response · Author response to Decision Letter 0]

2 Dec 2024

Response to Reviewers

Manuscript Title: Validating Quality Standards in Palestinian Emergency Departments: An e-Delphi Survey Approach

Manuscript ID: PONE-D-24-25666

Date: 27/11/2024

Dear Naeem Mubarak,

Academic Editor / PLOS ONE

Dear Reviewers,

We sincerely thank you for your time and constructive feedback, which has significantly enhanced the quality of our manuscript. Below, we provide detailed responses to each comment raised by the editor and reviewers. All revisions are highlighted (track changes) in the manuscript for your convenience.

Responses to Editorial Comments

Comment 1: PLOS ONE's style requirements.

Response: The manuscript has been revised to meet the PLOS style requirements.

Comment 2: Please include a complete copy of the PLOS questionnaire on inclusivity in global research in your revised manuscript.

Response: We have included the full text of the PLOS inclusivity questionnaire as Supporting Information (S1 Attachment) in the manuscript.

Comment 3: Please provide additional details regarding participant consent.

Response: Additional details have been added to the methods section under the subheading Delphi Panel Recruitment and Sample (Page 8). The study did not involve minors. These details have also been added to the “Ethics Statement” field of the submission form.

Comment 4: Please include captions for your Supporting Information files at the end of your manuscript and update any in-text citations accordingly.

Response: Captions for all Supporting Information files have been added to the end of the manuscript. Additionally, all in-text citations have been updated to match these files, ensuring consistency.

Comment 5: Please review your reference list to ensure that it is complete and correct.

Response: The reference list has been reviewed and confirmed to be complete and accurate.

Responses to Reviewer #1 Comments

Comment 1: Introduction – Elaboration on previous shortcomings in the literature.

Response: A paragraph has been added to the introduction on Page 5 / line 127, discussing conventional methods and the approach utilized in this study.

Comment 2: Methods – Delphi panel recruitment and sample.

Response: Measures to mitigate potential bias during Delphi rounds have been detailed under the Delphi Panel Recruitment and Sample section on Page 7 / line 168. Additionally, participants were allowed to complete the survey in stages using the Lime Survey platform to ensure thoughtful responses. Summarized the mitigation action as follows:

(Measures were taken at several stages to mitigate potential bias related to survey length. First, Previous efforts aimed to streamline standards by identifying appropriate ones for the Palestinian context through literature review and input from local experts, enhancing the clarity of the standards and aligning them with the study objectives. Second, Participants received clear instructions and guidance before starting the survey, including its purpose and estimated completion time. This transparency helped manage expectations and encouraged thoughtful responses. Additionally, reminders were sent during the survey period to encourage participation without exerting pressure, which could have compromised the quality of responses. Finally, the survey design through the Lime Survey platform allowed for completing the survey in several stages and not at the same time, as the participant can save the answers at the last point and complete it at another time if he feels bored, tired, or busy).

Comment 3: Methods – Readability, clarity, and comprehensiveness.

Response: A short paragraph is added within the methods section/study instrument/on page 8 / line 199.

Comment 4: Methods – Round 1 evaluation and shortlisting.

Response: The paragraph under Round 1 Evaluation and Shortlisting (Page 9 / line 219) has been rephrased, with added elaboration on the interquartile range (IQR).

Comment 5: Discussion – Literature comparison.

Response: The second paragraph of the discussion on (Page 14 / lines 331-340) has been rewritten to include a comparison with existing literature and explain the contribution of these criteria to ongoing improvements.

Comment 6: Literature Review – Examples of standards.

Response: Examples of standards from prior studies and literature have been included in the introduction on (Page 4/ lines 107 -118).

Comment 7: Formatting adjustments.

• Inclusion of line numbers: Resolved.

• Consistent page numbering format: Resolved.

• Renaming "Background" to "Introduction": Resolved.

Responses to Reviewer #2 Comments

Comment 1: The abstract is too long.

Response: The abstract has been shortened, particularly the methodology and conclusion sections.

Comment 2: Provide more details about Palestine's previously derived EDQS in the introduction.

Response: Additional details have been added on Page 7, Lines 138–149.

Comment 3: Trauma quality standards should have priority.

Response: These EDQS are comprehensive, covering all emergency services, including standards specific to trauma (e.g., equipment, resuscitation room, ACLS/BLS training). Details are provided in Appendix S 5.

Comment 4: 103 quality standards for Palestinian EDs are too many; fewer would be better.

Response: The final list comprises 100 validated EDQS. While comprehensive, the standards are divided into manageable domains, subdomains, and standards. Future research may explore their application to identify areas for consolidation.

We thank the reviewers and editor again for their valuable comments. Please do not hesitate to contact me if further clarification is needed. We look forward to your feedback.

Sincerely,

Abed Alra'oof Bani Odeh

UCT – PhD candidate

27/11/2024

---

## [Decision Letter · Decision Letter 1]

16 Dec 2024

Validating quality standards in Palestinian emergency departments: an e-Delphi survey approach

PONE-D-24-25666R1

Dear Dr. Abed Alra'oof Saleem Bani Odeh,

We’re pleased to inform you that your manuscript has been judged scientifically suitable for publication and will be formally accepted for publication once it meets all outstanding technical requirements.

Kind regards,

Naeem Mubarak, PhD

Academic Editor

PLOS ONE

Additional Editor Comments (optional):

The authors have convincingly addressed all the comments. No further changes are needed. Best of luck with your publication

Reviewers' comments:

Reviewer's Responses to Questions

**Comments to the Author**

1. If the authors have adequately addressed your comments raised in a previous round of review and you feel that this manuscript is now acceptable for publication, you may indicate that here to bypass the “Comments to the Author” section, enter your conflict of interest statement in the “Confidential to Editor” section, and submit your "Accept" recommendation.

Reviewer #1: All comments have been addressed

Reviewer #2: All comments have been addressed

2. Is the manuscript technically sound, and do the data support the conclusions?

Reviewer #1: Yes

Reviewer #2: Yes

3. Has the statistical analysis been performed appropriately and rigorously? 

Reviewer #1: Yes

Reviewer #2: N/A

4. Have the authors made all data underlying the findings in their manuscript fully available?

Reviewer #1: Yes

Reviewer #2: Yes

5. Is the manuscript presented in an intelligible fashion and written in standard English?

Reviewer #1: Yes

Reviewer #2: Yes

6. Review Comments to the Author

Reviewer #1: All the comments have been addressed by the authors, and I have no further suggestions to make. The modifications introduced are commendable, as they enhance clarity, address prior limitations, and provide thorough elaboration while acknowledging previous findings. The authors have demonstrated significant dedication to addressing feedback, particularly in advancing the objective of validating Emergency Department Quality Standards (EDQS) in Palestine using a unique approach.

Reviewer #2: The comments have been responded well to reach standard of worth of Publishing in PLOS One. The study is going to improve quality standards in Palestinian emergency departments

7. PLOS authors have the option to publish the peer review history of their article (what does this mean?). If published, this will include your full peer review and any attached files.

Reviewer #1: No

Reviewer #2: **Yes: **Imtiaz Wani

---

## [Editor Report · Acceptance letter]

28 Dec 2024

PONE-D-24-25666R1 

PLOS ONE

Dear Dr. Bani Odeh, 

I'm pleased to inform you that your manuscript has been deemed suitable for publication in PLOS ONE. Congratulations! Your manuscript is now being handed over to our production team.

Kind regards, 

on behalf of

Dr Naeem Mubarak 

Academic Editor

PLOS ONE